# Unexpected Gene-Flow in Urban Environments: The Example of the European Hedgehog

**DOI:** 10.3390/ani10122315

**Published:** 2020-12-07

**Authors:** Leon M. F. Barthel, Dana Wehner, Anke Schmidt, Anne Berger, Heribert Hofer, Jörns Fickel

**Affiliations:** 1Department of Evolutionary Ecology, Leibniz Institute for Zoo and Wildlife Research (IZW), Alfred-Kowalke-Straße 17, 10315 Berlin, Germany; Berger@izw-berlin.de; 2Berlin Brandenburg Institute of Advanced Biodiversity Research (BBIB), 14195 Berlin, Germany; hofer@izw-berlin.de; 3Department of Biology, Chemistry, Pharmacy, Freie Universität Berlin, Takustrasse 3, 14195 Berlin, Germany; Wehner.Dana@gmail.com; 4Department of Evolutionary Genetics, Leibniz Institute for Zoo and Wildlife Research (IZW), Alfred-Kowalke-Straße 17, 10315 Berlin, Germany; Aschmidt@IZW-Berlin.de; 5Department of Ecological Dynamics, Leibniz Institute for Zoo and Wildlife Research, Alfred-Kowalke-Straße 17, 10315 Berlin, Germany; 6Institute for Biochemistry & Biology, University of Potsdam, 14476 Potsdam-Golm, Germany

**Keywords:** urban, hedgehog, genetic cluster, barrier

## Abstract

**Simple Summary:**

An urban environment holds many barriers for mammals with limited mobility such as hedgehogs. These barriers appear often unsurmountable (e.g., rivers, highways, fences) and thus hinder contact between hedgehogs, leading to genetic isolation. In our study we tested whether these barriers affect the hedgehog population of urban Berlin, Germany. As Berlin has many of these barriers, we were expecting a strong genetic differentiation among hedgehog populations. However, when we looked at unrelated individuals, we did not see genetic differentiation among populations. The latter was only detected when we included related individuals too, a ‘family clan’ structure that is referred to as gamodemes. We conclude that the high percentage of greenery in Berlin provides sufficient habitat for hedgehogs to maintain connectivity across the city.

**Abstract:**

We use the European hedgehog (*Erinaceus europaeus*), a mammal with limited mobility, as a model species to study whether the structural matrix of the urban environment has an influence on population genetic structure of such species in the city of Berlin (Germany). Using ten established microsatellite loci we genotyped 143 hedgehogs from numerous sites throughout Berlin. Inclusion of all individuals in the cluster analysis yielded three genetic clusters, likely reflecting spatial associations of kin (larger family groups, known as gamodemes). To examine the potential bias in the cluster analysis caused by closely related individuals, we determined all pairwise relationships and excluded close relatives before repeating the cluster analysis. For this data subset (*N* = 65) both clustering algorithms applied (Structure, Baps) indicated the presence of a single genetic cluster. These results suggest that the high proportion of green patches in the city of Berlin provides numerous steppingstone habitats potentially linking local subpopulations. Alternatively, translocation of individuals across the city by hedgehog rescue facilities may also explain the existence of only a single cluster. We therefore propose that information about management activities such as releases by animal rescue centres should include location data (as exactly as possible) regarding both the collection and the release site, which can then be used in population genetic studies.

## 1. Introduction

Urbanisation involves some of the most rapid and intense human-induced transformation processes. Structures such as impervious surfaces, roads and buildings have fragmented the environment for many species. Semi-penetrable or impenetrable barriers now separate smaller patches of the former landscape, particularly in the urban conurbation. In order to gain access to adequate resources, animals living in such patches often have to cross these barriers to move between patches. Some wildlife species easily surmount barriers and cope with urban conditions (urban utilisers, urban dwellers; [1]) and the close proximity to people (e.g., wild boar *Sus scrofa*; [2]), whereas others cannot (e.g., great bustard *Otis tarda*; [3]). In addition, some species benefit from structures in urban spaces that mimic their original habitat (e.g., common swift *Apus apus*; [4]). Thus, for behaviourally flexible wildlife species urban habitats may provide a novel living environment with the opportunity to exploit novel resources [2,5,6].

Geographic separation of populations by barriers reduces gene-flow among them and thus increases genetic differentiation among populations. It also decreases the genetic variation within populations both by genetic drift and by reducing the availability of genetically different breeding partners, thereby increasing the risk of inbreeding and subsequent inbreeding depression as well as a higher incidence of infectious diseases and thus elevated mortality [7]. Thus, a consequence of habitat fragmentation may be local population extinction [8,9,10,11].

Urban landscapes also often contain large green patches such as parks, residential gardens, cemeteries, currently unused former industrial sites (brownfield sites) and other habitats that provide a relatively undisturbed living space for wildlife species. These patches may serve as stepping stone habitats, allowing gene flow between otherwise separated local populations [12]. Whether gene flow occurs depends on the mobility and dispersal capacity of each species in relation to the distances between suitable habitat patches and the distribution of the latter within the urban matrix. Thus, we expect species with high mobility and high dispersal capacity to be less affected by a strongly structured urban landscape (e.g., the red fox *Vulpes vulpes* [13]) than species with small home ranges and limited dispersal capacity. For the latter we therefore expect a fragmented urban landscape to promote genetic isolation of clusters of individuals, causing a highly structured meta-population [14,15,16,17].

The purpose of this study is to test these expectations by genotyping European hedgehogs (*Erinaceus europaeus*) across the highly structured and fragmented urban matrix of the city of Berlin, Germany. Although hedgehogs are widely distributed across Europe [18], we used this species as a model species because of its limited dispersal capacity and its relatively small home range [19,20]. The size of the latter may range from 0.8 ha (England; [21]), over 10 to 40 ha (England; [22]) up to 98 ha (Finland; [23]). Whereas female hedgehogs mostly stay within their habitat patches, male hedgehogs occasionally cover distances of up to 7 km per night [24]. Because the European hedgehog can use the urban matrix and cope with its structural characteristics [17,19], population densities in urban areas may actually be higher than in rural habitats [25,26]. However, despite their broad geographical distribution and their ability to utilise urban matrices, hedgehog populations have been declining in size and numbers across Europe [27,28,29,30,31]. Understanding the long-term consequences of progressive spatial fragmentation by urbanisation on hedgehog genetic population structure might become increasingly important for developing conservation strategies for this species [19,32,33].

## 2. Materials and Methods

### 2.1. Sample Collection and Sites

Over a period of five years (2013–2017), we collected mouth mucosal cells from free-ranging European hedgehogs (*N* = 250) using nylon swabs (FLOQSwabs, COPAN, Brescia, Italy) and Forensic cotton Swaps (Sarstedt, Nümbrecht, Germany) in the city of Berlin and its suburbs (~876 km^2^, or 87,594 ha). Sampling was carried out between 10 p.m. and 4 a.m. during torchlight transect walks in different public parks, cemeteries and green areas in Berlin. Due to the very low sampling success in the South-Western part of Berlin, we shifted the sampling effort during 2016 and 2017 to the North-eastern part of the city, mainly to two large parks: (a) the ”Treptower Park”, a 20 ha public park open to and accessible by the general public throughout day or night in south-central Berlin and (b) the ~160 ha large “Tierpark Berlin”, Europe’s largest landscape zoological garden and inaccessible to the general public after dusk. For each individual, we recorded the GPS coordinates of its sampling location. Additional samples (*N* = 56) were provided by animal rescue facilities and local veterinary surgeries in Berlin (Figure 1, Appendix A). For these samples approximate locations were provided by staff members. Coordinates, which may have had an error margin of a few hundred metres, where recorded using online maps. We also asked staff working at these facilities whether they had implemented particular rules on how release sites were chosen after the rehabilitation of hedgehogs. All procedures in this study involving animals were performed in accordance with the ethical standards of the institution (IZW permit 2016-02-01) and German federal law (permission numbers Reg0115/15 and G0104/14).

### 2.2. DNA Extraction and Analysis of Microsatellite Loci

DNA was extracted from all 306 samples using the DNeasy kit (Qiagen, Hilden, Germany) following the manufacturer’s instructions, with a final DNA-elution in 80 µL distilled water (sterile). DNA concentrations were measured spectrophotometrically using a NanoDrop1000 (PeqLab GmbH, Erlangen, Germany). Individuals were genotyped at 10 microsatellite loci using a panel of nine loci from a previous landscape genetics study [34], with locus EEU1 added. The panel consisted of the following loci: EEU1, EEU2, EEU3, EEU4, EEU5 and EEU6 [35], EEU12H, EEU37H, EEU43H, and EEU54H [36]. One primer per pair was 5′- labelled with a fluorescent dye (6-FAM or HEX). To save time and costs, we prepared (after optimization) four primer master-mixes (Mix-A to Mix-D, 50 µL each). Mix-A contained the primers for loci EEU1, EEU2, and EEU54H (all 1 µM), Mix-B consisted of primers for loci EEU6 (1 µM) and EEU12H (2 µM), Mix-C of primers for loci EEU3 (1 µM) and EEU37H (2 µM), and Mix-D included the primers for loci EEU4 (4 µM) and EEU5 (2 µM). Primer pair EEU43H (3 µM) was run separately. The genotyping PCR mixture (10 µL) consisted of 5 µL 2′Type-it^TM^ multiplex mix (Qiagen, Hilden, Germany), 1 µL primer mix, 3 µL H_2_O and 1 µL DNA (50–120 ng). Cycling conditions were equal for all four master-mixes and locus EEU43H and were performed as touchdown-PCR: 95 °C 5 min, 4′ {94 °C 30 s, 63 °C down to 57 °C in 2 °C increments of 90 s each, 72 °C 30 s}, 31’ {94 °C 30 s, 55 °C 90 s, 72 °C 30 s}, 60 °C 30 min final elongation. Amplification products were analysed by capillary electrophoresis on an A3130*xl* automated sequencer (Thermo Fisher, Waltham, MA, USA) using POP7 and sized by comparison to a Genescan™ 500 ROX™ Size Standard (ABI) using the software Genemapper v.3.7 following the manufacturer’s instructions. To avoid misleading results by allelic dropouts and false alleles, we applied a maximum likelihood approach [37] and genotyped each sample twice (in duplicates). The following quality filters were applied: (i) we did not allow for any allele mismatch between duplicates. If there was a mismatch, the sample was removed and genotyped again in duplicates from freshly extracted DNA. Genotypes were only scored if no mismatch was detected; otherwise, the sample was excluded from further analysis; (ii) we also excluded all individuals for which more than one locus had missing data.

### 2.3. Data Analysis

We calculated observed (*H*_O_) and expected heterozygosities (*H*_E_), number of alleles (*N*_A_), as well as potential deviations from Hardy-Weinberg equilibrium (HWE) using the program Cervus v.3.0.7 [38,39,40]). We also used Cervus to search for matching genotypes across all samples. Tests for the presence of genotypic disequilibria among loci were performed using the software package Arlequin v.3.5.2.2 [41,42]. The significance level α was Bonferroni-corrected and set at 0.001 (0.05: 45 pairwise comparisons). Potential presence of null alleles was assessed using MicroChecker v.2.2.3 [43].

Although hedgehogs are solitary animals, their limited dispersal capacity (compared with larger mammals) may cause a population genetic structure by which closely related individuals may be living in closer proximity to each other than to more unrelated individuals. Because some clustering algorithms are affected by such associations of kin [44], we determined pairwise relatedness (*r*, [45] among all samples using the software package Coancestry v.1.0.1.9 [46]. Pairs with *r* > 0.5 were marked and subsequent cluster analysis (see below) was performed with and without these pairs (Appendix A).

The possible presence of genotypic clusters was evaluated both for the subset of only unrelated individuals and for the whole data set. For this purpose, we used two software packages with a Bayesian clustering approach: Structure v.2.3.4 [47,48,49] and Baps v.6.0 [50,51]. As priors for Structure, we applied the admixture model in conjunction with the correlated allele frequency model, because it is better suited to detect a subtle population structure, although this makes it more likely to overestimate the number of clusters *K* [48]. The model was applied to *K*-values ranging from *K* = 1 to 8. The required allele frequency distribution parameter λ was estimated for each run. To determine both appropriate burn-in and Markov chain lengths for parameter estimates of allele frequencies and membership coefficients per genotype in each genotypic cluster (*Q*), we set *K* = 1 and watched for the likelihoods to converge under various burn-in and run lengths. The final burn-in length was set at 20,000 iterations and Markov chains were run with a length of 200,000 iterations each. Each *K* was assessed independently 10 times to verify the consistency of estimates across runs. The most likely *K* was determined using both the log likelihood values (as Δ*K* cannot be applied if *K* = 1) and by following the Δ*K* method [52] using Structure Harvester [53]. For Baps the *K* prior ranged from 2 to 8 (as Baps cannot detect *K* = 1), whereby each *K* was also independently assessed 10 times. In addition, we used a location prior by providing the GPS coordinates of each sample’s origin. We applied the algorithm using both the ‘admixture’ and the ‘no admixture’ prior.

### 2.4. Assignment

The threshold for the *Q*-value above which an individual will be assigned to a cluster is of importance. If the threshold is set too high it may underestimate a structure that in reality exists, whereas a threshold which is set too low will overemphasise a structure that in reality is not as pronounced as assumed. Here we chose a relatively conservative value of *Q* ≥ 0.85 as the threshold for the assignment of individuals [54,55], thus allowing for some gene flow to have occurred among the inferred ancestral populations (at least three generations ago). Genetic distances between clusters of assigned individuals as well as the number of migrants (*N*_m_) among clusters were estimated using Arlequin. Input files for the different programmes were generated using the software Create [56]. Assignment results were averaged over ten runs using R [57] as means ± standard deviations (SD) unless stated otherwise.

## 3. Results

### 3.1. Genotyping

#### 3.1.1. All Individuals

From the original dataset of 306 hedgehogs, data from 156 individuals had to be excluded. Out of those 156, 154 individuals were excluded because genotyping of their samples failed at more than one locus and two individuals were removed because the alleles of their duplicate sample genotypes did not match at all loci. In total, 150 individuals (49%) were successfully genotyped at all ten loci. Out of these genotypic profiles, one profile occurred three times and five others twice, leaving 143 unique genotypes (Appendix A). These 143 genotypes, however, were not evenly distributed across the city of Berlin. The reasons were the low sampling success in the southern parts of the city, the shifted focus of sampling efforts in 2016 and 2017, and the fact that many samples from the southern part of Berlin did not pass the quality control filters.

The number of alleles per locus (*N*_A_) ranged from four (locus EEU12H) to 16 (EEU37H), with a mean of 10.9 ± 4.1 (Table 1). *H*_O_ across all 143 unique genotypes ranged from 0.350 at locus EEU6 to 0.754 at locus EEU3, with a mean of *H*_O_ = 0.621 ± 0.133 (Table 1). Across all loci and individuals, one locus (EEH37H) deviated significantly from HWE (Table 1). Although several loci indicated the potential presence of null alleles, the probability was generally low. Pairwise relatedness analysis revealed numerous pairs of individuals with a high relatedness index (*r* ≥ 0.5). Removal of these related individuals reduced the data set to 65 unrelated hedgehogs (Appendix A). Presence of linkage disequilibria (*LD*) among the ten loci was tested both for the unrelated individuals (*N* = 65) and for 143 hedgehogs (all individuals). Among the unrelated individuals one out of 45 pairwise comparisons among the ten loci and among all hedgehogs 12 pairwise comparisons showed LD, although all loci had previously been declared to be independently inherited [36,58]; the former also included the loci from [34,58]. In our study, hedgehogs were sampled over a very large area (Figure 1), likely violating the assumption of an unstructured population, as expected for a small mammal in a highly fragmented landscape. The deviation from HWE at locus EEU37H and the linkage disequilibria may thus have been due to the Wahlund effect [59]. We therefore searched for an underlying population structure, first among the unrelated individuals and then among all individuals.

#### 3.1.2. Unrelated and Related Individuals

For the unrelated individuals data set, all individuals were assigned to a single cluster (mean LnP(*K*) = −2268.81 ± 0.481; results from Structure). For the entire data set, both clustering algorithms (Structure, Baps) indicated the presence of three to four genotypic clusters (Table 2). The Δ*K* estimate (Structure Harvester) favoured three clusters over four (Δ*K* for *K* = 3 was 52.25; Δ*K* for *K* = 4 was 51.4), whereas Baps favoured the presence of four clusters, with the fourth cluster being represented by two individuals (sampled at the same location). The likelihood for the number of genotypic clusters (*K*) to reflect the true number of ancestral populations had the following values (derived from Baps): for *K* = 3: 0.00136, *K* = 4: 0.98883, and for *K* = 5: 0.0098).

Using a value of *Q* ≥ 0.85 (Structure), 74 out of 143 genotypes (51.7%) were assigned to either one of three genotypic clusters: cluster 1 with 29 genotypes, cluster 2 with 14 genotypes (all individuals but one were from “Tierpark”), and cluster 3 with 31 genotypes (all but one from “Treptower Park”). The 69 remaining genotypes were admixed, with admixture occurring across all clusters (Table 2, Appendix A). Each cluster was at HWE. Observed (*H*_O_) and expected heterozygosities (*H*_E_) were *H*_O_ = 0.623 and *H*_E_ = 0.685 for cluster 1 (*N* = 29), for cluster 2 (*N* = 14) they were *H*_O_ = 0.557 and *H*_E_ = 0.524, and for cluster 3 (*N* = 31) they were *H*_O_ = 0.578 and *H*_E_ = 0.651. Pairwise genetic distances (*F*_ST_) among all clusters were significant (*p* < 0.05) with *F*_ST_ = 0.169 between clusters 1 and 2, *F*_ST_ = 0.11 between clusters 1 and 3, and *F*_ST_ = 0.192 between clusters 2 and 3.

Using the assignment threshold of *Q* ≥ 0.65 [60], the number of hedgehogs per cluster would increase by 45 for cluster 1 (*N*_(*Q*=0.65)_ = 74), by four for cluster 2 (*N*_(*Q*=0.65)_ = 18), and by three for cluster 3 (*N*_(*Q*=0.65)_ = 34). Increasing the number of individuals per cluster in such a way would at the same time reduce the number of admixed individuals considerably (*N*_(*Q*=0.65)_ = 17).

#### 3.1.3. Migrants

The number of migrants (*N*_m_) per generation also differed among the three clusters. They were *N*_m_ = 1.22 between clusters 1 (wide-spread) and 2 (“Tierpark”), *N*_m_ = 2.02 between clusters 1 and 3 (“Treptower Park”) and *N*_m_ = 1.05 between clusters 2 and 3. Applying the Baps clustering algorithm led to results very similar to the ones obtained from the Structure analysis, except for the introduction of a fourth cluster (2 individuals only) and an increase in the number of hedgehogs assigned to any cluster (Table 2). This increase in the number of individuals assigned to a cluster was particularly pronounced in cluster 1, into which Structure had only assigned 29 hedgehogs, whereas the Baps algorithm assigned three times as many individuals to that cluster (*N* = 87). Following the Baps assignment, hedgehogs from cluster 1 were also present in the “Tierpark” and the “Treptower Park”.

### 3.2. Release of Hedgehogs after Rehabilitation

Although rescue facilities had no particular rules regarding the selection of release sites for rehabilitated hedgehogs, general policy was to release hedgehogs into favourable habitats, independent of their point of geographic origin. This policy led to the release of hedgehogs at distances far from the facilities and far from their previous pick-up points, in some cases at distances of >100 km.

## 4. Discussion

Considering only unrelated individuals, hedgehogs were assigned to a single cluster, whose members were spread across the city (Figure 2). Such a lack of genetic population structure was surprising in light of the presence of many potential barriers, and it contrasts with results from a genetic study on 149 urban hedgehogs in the city of Zurich (Switzerland), where a strong differentiation had been observed across an area of ~10,000 ha [60]. There, despite an eight times smaller spatial scale than in Berlin, three genotypic clusters had been inferred. The Zurich hedgehog clusters were well delineated by a major inner-city transportation axis as an anthropogenic barrier and two rivers as natural barriers [60]. The authors concluded that urban green areas were the most suitable habitat type to facilitate gene flow, whereas all other land cover types were more likely to impede gene flow [60].

The Zurich study differed from ours in several aspects: Their threshold for assigning individuals to a genetic cluster was considerably lower (*Q* ≥ 0.65 instead of *Q* ≥ 0.85), and they did not consider the potential effect of association of kin on genetic population structure. In our study, unrelated individuals did not demonstrate any obvious population genetic structure, although the city of Berlin is much larger than Zurich and even more divided by several highways and large rivers or canals.

The inclusion of all individuals in our study indicated the presence of at least three genotypic clusters (*Q* ≥ 0.85), two of which were spatially well delineated (Structure: clusters 2 and 3, the “Tierpark” and the “Treptower Park”). If we used the threshold of *Q* ≥ 0.65 as in the Zurich study [60], the relatively low assignment threshold and the inclusion of all 143 individuals (without removal of closely related individuals) would have led us to the conclusion of a strong population genetic structure consisting of three well-delineated clusters. Using a higher assignment threshold and removal of related individuals will reduce the number of individuals that can be assigned to a genotypic cluster and thus has a strong influence on the number of individuals per cluster and on the number of potential clusters to be detected.

As the genetic structure in Berlin hedgehogs only appeared if related individuals were included in the cluster analysis, we suggest the differentiation detected here to be a reflection of an underlying kinship network of gamodemes rather than to be a reflection of allele frequencies of three ancestral populations. The emergence of such gamodemes may be facilitated by the fact that hedgehogs are promiscuous as well as philopatric and that they have hetero-paternal superfecundation [61]. We do not know whether hedgehogs differentiate between kin and non-kin during mating season [61], but a lack of such differentiation may also contribute to the emergence and maintenance of “gamodemes”. Such a gamodeme structure would also explain the local concentration of ‘cluster 2 individuals’ in the “Tierpark” and of “cluster 3 individuals” in the “Treptower Park”. Although the “Tierpark” is a large park-like area (~160 ha) that was preserved after World War II and established as a zoological garden in 1954, it is almost fully fenced and surrounded by big streets both in the north and the west and by railway tracks in the east and the south. Thus, gene flow between hedgehogs from the “Tierpark” and the surrounding areas is clearly restricted, explaining the confinement from hedgehogs of cluster 2 to the “Tierpark”. This is also shown by the significant pairwise *F*_ST_ values, which were the highest between clusters 2 and 3 (*F*_ST_ = 0.192) and clusters 2 and 1 (*F*_ST_ = 0.169). Interestingly, the hedgehogs inhabiting the “Treptower Park” (cluster 3) are strongly differentiated from the ones living in the “Tierpark” (*F*_ST_ = 0.192, lowest migration rate with *N*_m_ = 1.05), but not as strongly from the wide-spread cluster 1 (*F*_ST_ = 0.11, highest migration rate with *N*_m_ = 2.02). The main difference to the location “Tierpark” is that the “Treptower Park” is not fenced in and always accessible. However, it is bordered on one side by the river Spree (Berlin’s main river) and on its three other sides by heavy-traffic roads. An additional heavy-traffic road crosses the park longitudinally. Yet these barriers appear still to be more penetrable for hedgehogs than those in the “Tierpark”. The reason why these “park gamodemes” have become so large may be the low landscape resistance within the parks, whereas at the borders of the parks landscape resistance increases drastically.

Expected heterozygosity (*H*_E_) for individuals of the most wide-spread cluster (cluster 1: *H*_E_ = 0.685; *N* = 29) was even slightly higher than the value of *H*_E_ = 0.68 measured in a country-wide study in the Czech Republic (average sampling site distances >450 km; [34]), indicating that in Berlin the urban environment does not lead to a reduction of genetic variability in hedgehogs. Because individuals from clusters 2 and 3 of our study were confined to single parks, either to the “Tierpark” or to the “Treptower Park”, we expected to detect low observed (*H*_O_) and expected heterozygosities there. Even though the values were indeed lower than in the widespread cluster 1, they were only lower by a small margin (cluster 2: *H*_O_ = 0.557/*H*_E_ = 0.524; cluster 3: *H*_O_ = 0.578/*H*_E_ = 0.651) and even slightly higher than those measured in free ranging hedgehog populations from New Zealand affected by a founder effect (*H*_O_ = 0.42–0.52/*H*_E_ = 0.51–0.57; [62]). These values were also in the range of values for the three urban clusters detected in Zurich (*H*_O_/*H*_E_ = 0.6/0.605; 0.523/0.568; 0.631/0.627), although pairwise population *F*_ST_-values among the Zurich clusters ranged only between 0.059 and 0.082 [60], indicating a higher proportion of shared alleles among the Zurich hedgehog clusters than among the gamodeme clusters of Berlin. The values measured in the urban hedgehogs of Berlin were also much higher than the ones found in a recent study on rural hedgehogs in Denmark [61]. Here the population (*N* = 14) of Bornholm Island had values of only *H*_O_ = 0.124 and of *H*_E_ = 126, while the highest values were recorded in a subpopulation of Jutland (south of the Limfjord, *N* = 71) with *H*_O_ = 0.293 and of *H*_E_ = 0.318.

In contrast to many other mammalian species, hedgehogs lack a clear dispersal phase [63,64]. They rarely cover distances larger than 4 km [63] and are restricted in their movements by roads and other barrier-like structural elements [65,66]. Because the city of Berlin is approximately eight times larger than Zurich, the emergence of a genetic population structure due to restricted gene flow as seen in Zurich (but see our comments above on Q) appears to be inevitable. Yet hedgehogs in Berlin did not differentiate into a clear population genetic structure (if related animals were excluded), although the city of Berlin is much larger than Zurich. We thus expected dispersal over these large within-city distances to be even less likely (than in Zurich) and therefore a genetic population structure to be even more pronounced and clearly delineated by space. This was, however, not the case. Our results and observations would be compatible with the idea that all Berlin hedgehogs derived from a single ancestral population.

Because our results provide only a temporal snapshot, we do not know whether the spatial discrimination of clusters 2 (“Tierpark”) and 3 (“Treptower Park”) is the beginning of a process leading either to population differentiation or to complete admixture (as we found numerous admixed individuals), or whether it may represent a stable genotypic equilibrium.

Although we currently do not have a detailed knowledge about the ancestry of hedgehogs in Berlin, it is well known that hedgehogs have lived in Berlin for centuries and have experienced Berlin’s increasing urbanisation throughout this period [67]. This raises the question as to what could be the reasons for the lack of a clear, spatially derived population genetic structure in a species that is considered to be substantially constrained by physical urban structures such as waterways, motorways, railways, and built-up areas [20,63,65], structures that characterise Berlin.

We argue that the main reason for our finding is the large proportion of green areas in Berlin. The city of Berlin is covered by 15,752 ha of forests (18%) and 10,885 ha of public green sites (12.4%) such as cemeteries, parks and gardens [2,68]. These areas provide a connective web of suitable habitats within the urban matrix, improving the opportunities for hedgehogs to maintain some amount of gene flow across the city. In addition, or alternatively, other factors may increase admixture. Given home ranges of 10–40 ha [22], the distances that needed to be covered to establish gene flow between “gamodeme” clusters are quite large for a short-legged ground-dwelling species, but numerous small and larger green areas can be stepping stones to link distant parts of the city. We also suggest that admixture had been enhanced by animals released by hedgehog rescue facilities [69]. These events are not fully quantified at present, but our interviews with personnel from rescue facilities confirmed that they are a regular occurrence, some estimates suggesting several hundred per year. Such rescue related translocations have also been observed in other studies [60,69,70].

## 5. Conclusions

We originally hypothesised urban hedgehogs, a species with relatively low mobility and low dispersal capacity, to be highly influenced by fragmented urban landscapes leading to genetic isolation of populations and thus a highly structured meta-population. Yet the hedgehog population in the city of Berlin is not genetically structured, if only unrelated individuals are being taken into account. A genetic structure becomes only visible if related individuals are also included in the analysis. Gene flow between these gamodeme-clusters is probably realised through natural means across the numerous green patches of Berlin’s urban matrix and complemented by anthropogenic translocations. To maintain the currently existing genetic diversity in Berlin’s hedgehog population, we suggest its repeated monitoring by census measures and population wide genetic analysis to determine whether current clusters (gamodemes) are at risk of becoming isolated.

## Figures and Tables

**Figure 1 animals-10-02315-f001:**
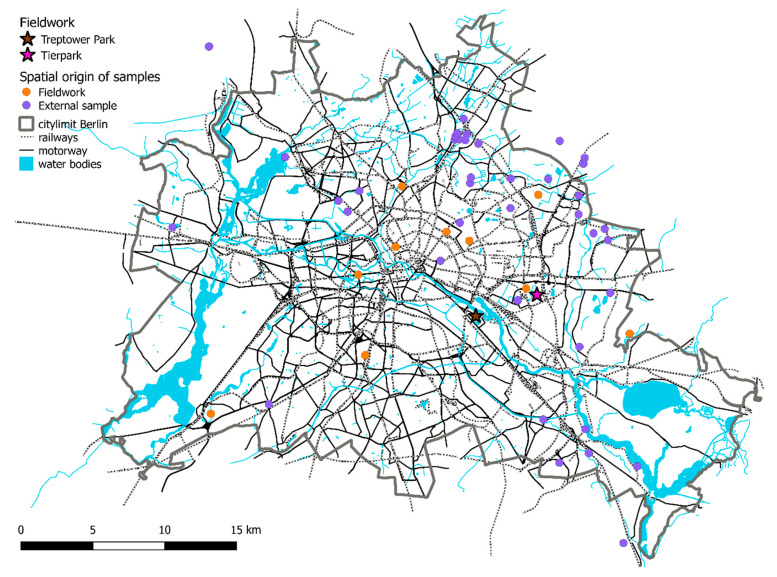
Map of Berlin and its surroundings and showing the locations from 139 out of 143 samples (four samples not shown because locations are outside of map). External samples: samples received from rescue centres. Their locations need to be viewed with caution. Samples from “Tierpark” and from‚ “Treptower Park” are lumped under pink and dark stars, respectively.

**Figure 2 animals-10-02315-f002:**
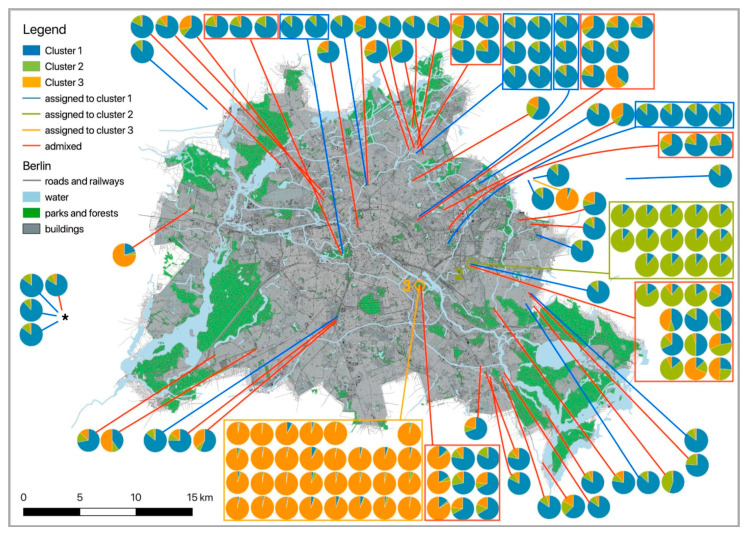
Map of Berlin and its surroundings indicating the locations of all samples displayed as pie charts showing the proportional membership of each individual in either one of three clusters (determined by Structure, including related animals). Different colours indicate membership to different clusters. *Q*-values were taken from Table 2. Individuals that belong to a cluster (*Q* ≥ 0.85) are grouped within squares framed in the colour of their cluster. Admixed individuals are framed in red squares. Solid lines connect groups and individuals with their sampling location. For individuals assigned to either one of the three clusters, colours of lines correspond to cluster membership (cluster 1: blue, cluster 2: bright green, cluster 3: bright orange). Red lines connect admixed individuals to their sampling site. Pie charts of the four samples that were collected outside of the displayed city area are given on the left side, indicated by *. Locations “Tierpark” and “Treptower Park” are indicated by coloured circles and cluster number (2, green: Tierpark; 3, orange: Treptower Park).

**Table 1 animals-10-02315-t001:** Indices of ten microsatellite loci across the 143 unique genotypes (upper part) and averaged for the clusters (lower part).

Locus	*N* _typed_	*N* _A_	Allele Size Range (bp)	*H* _O_	*H* _E_	HWE	*f* _Null_
EEU1	143	8	129–143	0.671	0.773	+	0.062
EEU2	141	13	257–281	0.752	0.863	+	0.064
EEU3	142	15	131–181	0.754	0.868	+	0.064
EEU4	143	14	144–170	0.699	0.785	+	0.052
EEU5	143	13	107–139	0.678	0.711	+	0.011
EEU6	143	6	145–159	0.350	0.331	+	−0.049
EEU12H	143	4	91–97	0.497	0.615	+	0.098
EEU37H	142	16	236–280	0.676	0.839	-	0.095
EEU43H	143	12	146–172	0.657	0.730	+	0.047
EEU54H	142	8	276–296	0.479	0.551	+	0.067
Mean SD	142.5 0.71	10.9 4.09		0.621 0.133	0.707 0.167		
Cluster assignment (*Q* > 0.85)		mean *N*_A_		*H* _O_	*H* _E_	HWE	
1	29	8.6		0.623	0.685	+	
2 (“Tierpark”)	14	3.3		0.557	0.524	+	
3 (“Trepower Park”)	31	6.0		0.578	0.651	+	
single cluster (*N* = 65)	65	10.6		0.672	0.731	+	
no cluster, all (*N* = 143)	143	10.9		0.621	0.706	- *	

*N*_typed_: number of individuals successfully genotyped at that locus, *N*_A_: number of alleles per locus, bp.: base pairs, *H*_O_: observed heterozygosity, *H*_E_: heterozygosity expected under HWE, HWE: Hardy-Weinberg equilibrium. (+): locus was at HWE, (-): locus deviated from HWE, *f*_Null_: probability for the presence of null-alleles (underlined values indicate an increased probability for the presence of null-alleles), SD: standard deviation. *: five out of 10 loci deviated from HWE.

**Table 2 animals-10-02315-t002:** Animal ID, sampling location, *Q*-values (STRUCTURE) and cluster assignment for 143 hedgehogs of Berlin.

Internal ID	Locality	STRUCTURE *Q*-Value for Cluster 1	STRUCTURE *Q*-Value for Cluster 2	STRUCTURE *Q*-Value for Cluster 3	STRUCTURE Cluster Assignment (*Q* ≥ 0.85)	BAPS Cluster Assignment *
147	Tiergarten, Berlin	0.87559	0.10548	0.01895	1	4
167	Tiergarten, Berlin	0.88068	0.10901	0.01031	1	4
176	Eisenhuettenstadt	0.85406	0.11966	0.02629	1	1
161	Hans-Baluschek-Park,	0.86127	0.11218	0.02655	1	1
311	Tierpark, Berlin	0.87213	0.10962	0.01824	1	1
334	12623 Berlin	0.86935	0.11628	0.01438	1	1
335	12623 Berlin	0.85235	0.11496	0.03271	1	1
220	Friedenstr., Berlin Near Volkspark Friedrichshain	0.87103	0.11977	0.00921	1	1
175	Volkspark Prenzlauerberg, Berlin	0.86135	0.105	0.03368	1	1
199	Hellersdorf, Berlin	0.86134	0.12691	0.01175	1	1
159	Prenzlauerberg	0.85732	0.10238	0.04031	1	1
338	Park am Weidengrund	0.87595	0.11224	0.01182	1	1
117	Near graveyard	0.88408	0.10591	0.01	1	1
326	Zum Erlenbruch, 15344 Strausberg	0.86423	0.11255	0.02321	1	1
156	Buergerpark Pankow-Berlin	0.86845	0.11301	0.01854	1	1
341	Kleingartenanlage 750 Jahre Berlin, 13057 Berlin	0.85329	0.10368	0.04302	1	1
328	Warnemuender Str. 18, 13059 Berlin	0.86432	0.11747	0.01821	1	1
329	Warnemuender Str. 18, 13059 Berlin	0.86724	0.1151	0.01767	1	1
330	Warnemuender Str. 18, 13059 Berlin	0.87485	0.1077	0.01746	1	1
337	KGA Maerchenland, 13089 Berlin	0.87728	0.10819	0.01456	1	1
231	Friedenstr. 8, 16356 Ahrensfelde	0.87273	0.10608	0.02116	1	1
257	Dietrichstr. 5, 16356 Ahrensfelde	0.87167	0.10645	0.0219	1	1
193	Jungbornstr., 13129 Berlin	0.85709	0.12218	0.02072	1	1
189	Strasse 7, 13129 Berlin	0.85836	0.11349	0.02814	1	1
194	Schwarzwaldstr./Ilsenstr., 13129 Berlin	0.85756	0.11789	0.02457	1	1
185	Gutenfelsstr. 14, 13129 Berlin	0.8736	0.11369	0.01271	1	1
187	Gutenfelsstr. 14, 13129 Berlin	0.8837	0.1066	0.0097	1	1
113	Choise-le-Roi-Str. 3, Berlin	0.88496	0.10623	0.00881	1	1
118	Vielitzsee Ortsteil Strubensee, 16835 green area	0.85436	0.13413	0.01153	1	1
179	Eisenhuettenstadt	0.84074	0.10226	0.05701	admixed	1
243	Zeuthen	0.6249	0.20766	0.16744	admixed	1
129	Rohrwallallee 10. 12527 Berlin	0.83142	0.10412	0.06446	admixed	1
120	Altglienike Feldweg	0.84766	0.10475	0.0476	admixed	1
137	Kablower Weg 89, 12526 Berlin	0.78369	0.0984	0.1179	admixed	1
138	Kablower Weg 89, 12526 Berlin	0.8267	0.1031	0.0702	admixed	1
125	Riesserseestr. 10. 12527 Berlin	0.84665	0.11413	0.03923	admixed	1
135	Korkedamm 73, 12524 Berlin	0.69588	0.08972	0.21436	admixed	1
127	Rehwiese, Gerkrathstraße 2 Park	0.72131	0.14963	0.12906	admixed	1
182	Zehlendorf, Berlin	0.40151	0.09898	0.49948	admixed	1
158	Hans-Baluschek-Park, 10829 Berlin	0.56664	0.07147	0.36189	admixed	1
169	Hans-Baluschek-Park, 10829 Berlin	0.75432	0.10163	0.14405	admixed	1
235	Glasberger Str. 43, 12555 Berlin	0.77728	0.10906	0.11368	admixed	1
110	Trainierbahn Hoppegarten	0.55419	0.42731	0.01849	admixed	1
A35_088	Treptower Park	0.66656	0.14738	0.18608	admixed	1
A4_317	Treptower Park	0.81876	0.16614	0.01512	admixed	1
A61_108	Treptower Park	0.67049	0.09899	0.23051	admixed	1
A68_108	Treptower Park	0.71856	0.17835	0.10308	admixed	1
126	Moldaustr. 30. 10319 Berlin near Tierpark	0.68485	0.2057	0.10944	admixed	1
128	Moldaustr. 24, 10319 Berlin near Tierpark	0.13048	0.18259	0.68692	admixed	1
136	Moldaustr. 24, 10319 Berlin near Tierpark	0.2581	0.25296	0.48897	admixed	1
174	Tierpark, Berlin	0.48697	0.40986	0.10316	admixed	1
308	Tierpark, Berlin	0.8438	0.13492	0.02127	admixed	1
310	Tierpark, Berlin	0.45013	0.09578	0.45409	admixed	1
314	Tierpark, Berlin	0.50102	0.45028	0.0487	admixed	1
317	Tierpark, Berlin	0.66602	0.15881	0.17519	admixed	1
320	Tierpark, Berlin	0.22576	0.45684	0.3174	admixed	1
333	12623 Berlin	0.76214	0.22126	0.01663	admixed	1
143	Tiergarten, Berlin	0.82986	0.15939	0.01076	admixed	1
152	Tiergarten, Berlin	0.79801	0.14577	0.05625	admixed	1
166	Tiergarten, Berlin	0.78003	0.16683	0.05316	admixed	1
309	Nordbahnhof park	0.72988	0.1093	0.16079	admixed	1
134	Volkspark Prenzlauerberg, Berlin	0.84225	0.1053	0.05244	admixed	1
142	Volkspark Prenzlauerberg, Berlin	0.62693	0.08785	0.28521	admixed	1
153	Volkspark Prenzlauerberg, Berlin	0.77372	0.0954	0.13087	admixed	1
168	Volkspark Prenzlauerberg, Berlin	0.75805	0.0945	0.14747	admixed	1
170	Volkspark Prenzlauerberg, Berlin	0.35675	0.0564	0.58687	admixed	1
172	Volkspark Prenzlauerberg, Berlin	0.83185	0.10633	0.06181	admixed	1
324	Eisenacher Str.,12629 Berlin near park	0.71554	0.09224	0.19223	admixed	1
300	Kastanienallee 122/126, 12627 Berlin near Teupitzer Park	0.83854	0.12196	0.03952	admixed	1
261	Wolfshorststr. 25, 13591 Berlin	0.20168	0.04025	0.75806	admixed	1
340	Mahlerstraße, 13088 Berlin	0.75765	0.09987	0.14248	admixed	1
241	Glambecker Ring 4, 12679 Berlin	0.66759	0.16157	0.17085	admixed	1
114	Togostr. 45, 13351 Berlin near Volkspark Rehberge	0.61322	0.10559	0.28119	admixed	1
248	13053 Berlin	0.58362	0.0891	0.32725	admixed	1
119	Ghanastr. 27, 13351 Berlin near Volkspark Rehberge	0.7999	0.09627	0.10382	admixed	1
139	Falkenberger Krugwiesen, 13057 Berlin	0.78143	0.11325	0.10533	admixed	1
140	Falkenberger Krugwiesen, 13057 Berlin	0.73941	0.18507	0.0755	admixed	1
150	Buergerpark Pankow-Berlin	0.66583	0.13902	0.19514	admixed	1
188	Schwarzelfenweg 19, 13088 Berlin	0.59525	0.23319	0.17157	admixed	1
116	Alt-Tegel 47c, 13507 Berlin	0.82319	0.14861	0.0282	admixed	1
191	Strasse 26 Nr. 30. 13129 Berlin near green area	0.65149	0.31845	0.03009	admixed	1
196	Schwarzwaldstr., 13129 Berlin	0.67859	0.10107	0.22037	admixed	1
186	Gutenfelsstr. 14, 13129 Berlin	0.81717	0.1626	0.02022	admixed	1
198	Gutenfelsstr. 14, 13129 Berlin	0.76845	0.1193	0.11227	admixed	1
200	Gutenfelsstr. 14, 13129 Berlin	0.8493	0.10802	0.04267	admixed	1
192	Urbacher Str., 13129 Berlin	0.54983	0.25383	0.19634	admixed	1
184	Freischuetzstr., 13129 Berlin	0.81933	0.14711	0.03358	admixed	1
203	Freischuetzstr., 13129 Berlin	0.8327	0.13544	0.03187	admixed	1
197	Krontalerstr., 13125 Berlin	0.84704	0.10899	0.04399	admixed	1
A3_317	Treptower Park	0.77842	0.11654	0.10505	admixed	admixed
A34_078	Treptower Park	0.67359	0.09012	0.23627	admixed	admixed
144	Tierpark, Berlin	0.10456	0.88882	0.00662	2	2
146	Tierpark, Berlin	0.10652	0.8851	0.00836	2	2
154	Tierpark, Berlin	0.10959	0.88179	0.00862	2	2
165	Tierpark, Berlin	0.10523	0.88487	0.00991	2	2
305	Tierpark, Berlin	0.10535	0.88925	0.00539	2	2
306	Tierpark, Berlin	0.10434	0.89067	0.005	2	2
307	Tierpark, Berlin	0.12173	0.85912	0.01916	2	2
312	Tierpark, Berlin	0.10548	0.88768	0.00684	2	2
313	Tierpark, Berlin	0.10518	0.88331	0.0115	2	2
315	Tierpark, Berlin	0.11634	0.86767	0.01601	2	2
318	Tierpark, Berlin	0.11617	0.87305	0.01078	2	2
319	Tierpark, Berlin	0.11028	0.87549	0.01424	2	2
342	Tierpark, Berlin	0.11259	0.87716	0.01028	2	2
344	IZW Garten, Berlin (bordering with Tierpark)	0.10948	0.87852	0.01199	2	2
141	Tierpark, Berlin	0.14033	0.79122	0.06844	admixed	2
149	Tierpark, Berlin	0.10468	0.79688	0.09844	admixed	2
321	Tierpark, Berlin	0.15174	0.82023	0.02802	admixed	2
322	Tierpark, Berlin	0.1618	0.8123	0.02591	admixed	2
157	Treptower Park	0.02233	0.01133	0.96636	3	3
345	Treptower Park	0.01293	0.01151	0.97556	3	3
346	Treptower Park	0.02053	0.00877	0.97068	3	3
348	Treptower Park	0.07228	0.0132	0.9145	3	3
349	Treptower Park	0.0197	0.00997	0.97034	3	3
350	Treptower Park	0.01276	0.01003	0.97721	3	3
A1_317	Treptower Park	0.01622	0.00786	0.97591	3	3
A10_028	Treptower Park	0.02804	0.02522	0.94674	3	3
A11_028	Treptower Park	0.06713	0.02515	0.90771	3	3
A12_028	Treptower Park	0.03142	0.0546	0.914	3	3
A13_028	Treptower Park	0.04447	0.01617	0.93935	3	3
A14_028	Treptower Park	0.01659	0.00572	0.97765	3	3
A15_028	Treptower Park	0.01453	0.01432	0.97112	3	3
A16_028	Treptower Park	0.04624	0.01015	0.94359	3	3
A2_317	Treptower Park	0.03028	0.01088	0.95885	3	3
A20_038	Treptower Park	0.01454	0.01811	0.96735	3	3
A21_038	Treptower Park	0.05401	0.01349	0.9325	3	3
A22_038	Treptower Park	0.04167	0.01371	0.94463	3	3
A25_078	Treptower Park	0.01262	0.02414	0.96325	3	3
A27_078	Treptower Park	0.02487	0.03366	0.94146	3	3
A28_078	Treptower Park	0.01631	0.01948	0.96424	3	3
A30_078	Treptower Park	0.02939	0.01268	0.95794	3	3
A31_078	Treptower Park	0.01652	0.04733	0.93614	3	3
A32_078	Treptower Park	0.01376	0.01006	0.97617	3	3
A37_088	Treptower Park	0.01249	0.01269	0.97483	3	3
A43_088	Treptower Park	0.02567	0.02771	0.94662	3	3
A47_098	Treptower Park	0.01487	0.01031	0.97481	3	3
A5_317	Treptower Park	0.02926	0.0273	0.94345	3	3
A59_108	Treptower Park	0.0119	0.00944	0.97865	3	3
A9_028	Treptower Park	0.00902	0.00582	0.98515	3	3
252	Friedenstr., 16356 Ahrensfelde	0.04766	0.01531	0.93704	3	3
A56_098	Treptower Park	0.12642	0.03009	0.84349	admixed	3
A62_108	Treptower Park	0.13874	0.02767	0.83359	admixed	3
343	Tierpark, Berlin	0.16607	0.05859	0.77535	admixed	admixed

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
