# Peer review of "Unexpected Gene-Flow in Urban Environments: The Example of the European Hedgehog"

_animals, 2020, doi:10.3390/ani10122315_

Round 1

Reviewer 1 Report

Dear authors,

Thank you very much for including all the comments and suggestions and the detailed answers applying each of them. 

I think the manuscript is much improved and I only have two small suggestions: 

1) On two occasions you write "City" with a capital C, should be city

2) Thanks for adding Figure 1. I think it adds a lot to the understanding. However, I think the map itself is very hard to see as there is a lot going on. Would it be possible to remove the streets on the map to see all the other lines better? Or otherwise you could increase the opacity of the background layer, to better be able to see the lines to the pie charts. 

Author Response

Response to Reviewer 1 Comments

Thank you very much for including all the comments and suggestions and the detailed answers applying each of them.

I think the manuscript is much improved and I only have two small suggestions:

1) On two occasions you write "City" with a capital C, should be city

Response:

Thank you. We changed the typos accordingly.

2) Thanks for adding Figure 1. I think it adds a lot to the understanding. However, I think the map itself is very hard to see as there is a lot going on. Would it be possible to remove the streets on the map to see all the other lines better? Or otherwise you could increase the opacity of the background layer, to better be able to see the lines to the pie charts.

Response:

Thank you for your suggestion. We guess that you meant Figure 2 instead of Figure 1 and we changed it as following: We decreased the opacity of the background map of Berlin and also changed the roads and railways from black to grey. Now, the lines to the pie charts are much better to see.

We did not want delete the roads and railways completely from the map as they (as well as the rivers and lakes) work as migration hinds for the hedgehogs.

Reviewer 2 Report

Dear authors. 

Thank you for providing a new, edited version of the manuscript and for taking my comments into consideration and responding to these in detail. 

I only have a few extra comments for you to consider: 

You have introduced the concept of gamodemes. Perhaps it would be useful for the readers if you defined this the first time you mention it? In the simple summary and abstract you still refer to “family clans”. Would it be an idea to introduce the term “gamodemes” here as well? E.g. in the simple summary you could add “a structure we call ‘family clan’ or “gamodemes”” to the text. 

In the beginning of the Results section you write: “These 143 genotypes, however, were not evenly distributed across the city of Berlin. The reasons were the low sampling success in the southern parts of the city, the shifted focus of sampling efforts in 2016 and 2017, and the fact that many samples from the southern part of Berlin did not pass the quality control filters.” Which quality control filters are you referring to? It seems this concept is not mentioned under “Methods”, at least not with the label “quality control filters”. 

I really enjoyed reading the manuscript and am very happy about the improved overview in Fig. 2. 

Author Response

Response to Reviewer 2 Comments

Dear authors.

Thank you for providing a new, edited version of the manuscript and for taking my comments into consideration and responding to these in detail.

I only have a few extra comments for you to consider:

You have introduced the concept of gamodemes. Perhaps it would be useful for the readers if you defined this the first time you mention it? In the simple summary and abstract you still refer to “family clans”. Would it be an idea to introduce the term “gamodemes” here as well? E.g. in the simple summary you could add “a structure we call ‘family clan’ or “gamodemes”” to the text.

Response:

It was actually a mistake from us, because in the final checkup prior to resubmitting the paper, we forgot to check the “simple summary” and only noticed our mistake after resubmission. Initially we only wanted to correct our mistake and just wanted to replace “family clans” with “gamodemes”. However, we immediately adopted the idea of the reviewer, because “family clan” is something that a “non-ecological” audience will immediately comprehend, while the term “gamodemes” addresses the “ecological” audience. Thus, we followed the suggestion made by the reviewer and kept both terms in the “simple summary”. We also added “gamodemes” in the “abstract” (… larger family groups, known as gamodemes…), while then using the term “gamodemes” only (without repetition of “family clan”) throughout the rest of the manuscript.

In the beginning of the Results section you write: “These 143 genotypes, however, were not evenly distributed across the city of Berlin. The reasons were the low sampling success in the southern parts of the city, the shifted focus of sampling efforts in 2016 and 2017, and the fact that many samples from the southern part of Berlin did not pass the quality control filters.” Which quality control filters are you referring to? It seems this concept is not mentioned under “Methods”, at least not with the label “quality control filters”.

Response:

The filtering we are referring to are the following two steps: i) each individual was genotyped twice and both genotypes had to be identical, and ii) we only allowed missing data per individual for one locus. We described this filtering approach in the Methods section, although without explicitly calling it “filtering”. In the manuscript we wrote:

“We did not allow for any allele mismatch between duplicates. If there was a mismatch, the sample was removed and genotyped again in duplicates from freshly extracted DNA. Genotypes were only scored if no mismatch was detected; otherwise, the sample was excluded from further analysis. We also excluded all individuals for which more than one locus had missing data. “

We have now added a short half-sentence to avoid confusion: The following quality filters were applied: i) we did not allow ….; ii) we also excluded …

I really enjoyed reading the manuscript and am very happy about the improved overview in Fig. 2.

Response:

We thank the reviewer for this assessment.

This manuscript is a resubmission of an earlier submission. The following is a list of the peer review reports and author responses from that submission.

Round 1

Reviewer 1 Report

This study examines the genetic structure of hedgehogs in Berlin using ten microsatellite loci. The cluster analysis including all individuals yielded three genetic clusters. When removing related individuals, the remaining unrelated individuals form one cluster.

This study is a valuable addition in this field, which needs more evidence and replication in other cities to show general patterns over Europe. I particularly liked the distinction between the clustering including related individuals and unrelated individuals. I think this is a good point to keep in mind.

However, there are some major issues that need to be addressed before publication.

The discussion contains good arguments that could be structured in a more consistent way.

The visualization of the clustering could be improved either by using structure bar plots, some visualization on a map, pie charts or a combination. Visually a plot may be more quickly informative even if the table is needed to provide detailed numbers.

Also while you give Ho and other statistics for the loci in table 1, it would be nice to also see this information in a table for the clusters you find for the unrelated individuals data set so it can be more easily compared to other studies. This information is in the text, but is not easily accessible.

Berlin is a city with a special history of fragmentation. I was surprised to not hear the Berlin Wall mentioned at all. Would you not have expected this to have an influence on the ability of hedgehogs to move?  

Further points:

37:           … all individuals in the cluster…

38 & 39: “potentially” is used in both sentences. Maybe replace it once with another word.

45:           This sentence ending seems grammatically wrong. Maybe replace with “for existence of only a single cluster.”

45:           This sentence is both vague and difficult to understand. What do you mean with data collection? Do you want to note down this information or should this individuals be sampled?

50:           “of the formerly existing landscape” is not necessary in this sentence.

58:           In addition, some species benefit from structures in urban spaces that mimic their original habitat (e.g. common swift).

86:           A paper analyzing the decline of urban hedgehogs (Taucher et al. 2020 in Animals) just came out, which might be useful to cite here, as you are particularly talking about urban spaces.

93:           Here the sample size is referred to 250 individuals, whereas on line 109 306 samples are mentioned. How does this come about?

95:           Here you are using “km2” on line 98 and later in the text you use “ha”. I would suggest using the same units throughout the manuscript.

99:           Maybe add “sampling” before location

172:         Here it would be good to cite some literature to support your choice in values.

177:         presented does not seem the right word choice here à analysed

182:         This sentence should be written more concisely.

182:         Is it normal to have to exclude so many samples? There are only very few samples remaining from the south of the city. Why do you have so few samples from there? You should discuss the distribution of your samples in a bit more detail.

199:         Is this shown in Figure 1? If so, refer to it.

209:         It would be great to have some visual help of the clustering on a map.

238:         rescue facilities

246:         Figure 1 and Figure 2 are nearly indistinguishable. I would suggest that you have one figure that is the map of all of your sampling and another which perhaps shows the clustering in a more visually appealing way (perhaps using pie charts).

255:         If kin relatedness might be the cause for the differences between Zurich and Berlin, as suggested by you, would you expect the Zurich clusters to have lower Ho than what you have? Overall, since a big part of the discussion is the comparison with the Zurich study, perhaps it would be worth to touch upon Zurich’s Ho and Fst. This could make your point stronger.

256:         It would be interesting to see what happens with your data if you use the Q used in Zurich. Have you tried this?

264:         Replace into with in

265:         Has the expression ‘family clans’ been put forward previously in the literature? If so, cite it.

281:         Were the Tierpark and the Treptower Park set up/colonized by hedgehogs approximately around the same time? You never talk about time since colonization. Would you not expect that to have an influence on the clustering you observe now?

294:         Maybe you could cite the paper on juvenile hedgehogs (Rasmussen et al. 2019) here.

296:         Brackets for references need to be fixed here.

299:         This sentence does not really fit in here. Mention the assumption before the previous sentence.

313:         Replace the question mark with a period.

316:         On lines 267 to 272 you describe how the two parks seem pretty isolated from the surrounding. Here you mention the connective web of parks and green spaces. Both points seem a bit contradictory and should be further explained.

525:         The legends for figure 1 and 2 should be fixed (some are capitalized, some are not). Make sure to be consistent.

545:         Table 2 seems quite unwieldy for the main text.

Reviewer 2 Report

The paper of Barhtel et al. represent important contribution to a knowledge of hedgehog landscape genetics in urban environment. As far as I can ascertain, the analytical part was conducted without substantial flaws.

I have following questions and comments:

1) Why the sex was not determined in investigated specimens? Given typical female philopatry in mammals, lack of this information could bias the results and should be at least mentioned in the Discussion section.

2) 182-185: In my opinion, setting the zero threshold for missing data and discarding 50% of genotypes in case of fresh tissue samples (buccal swabs) is an accuracy overkill and it is weakening statistical power of the study, but the decision is upon authors.

3) To detect potential subtle substructure, it would be useful to use also Structure analysis with Locprior option on. Have you tried to treat the potential subpopulations (i.e. samples separated by pronounced barriers) as candidate populations in Structure analysis?

Minor comments:

24, 34: Hedgehogs could not be categorized as “small mammals”, they belong to the category “medium-sized mammals” with pronounced evolutionary consequences (evolution of antipredator autapomorphies etc.). Given the context, use rather “mammals with limited mobility” or similar expression.

93: FLOQswabs are not from cotton, but from nylon.

30, 265, etc.: Maybe the biological term “deme” or “gamodeme” would be more suitable than rather undefined expression “family clan”.

The chapter 3.2 is not suitable for Result section and it should be rather moved to the Discussion. Please also note the recent paper focused to the issue of potential homogenization of hedgehog population structure related to treatment in wildlife shelters (Ploi et al. 2020). Are there any differences between Berlin and Zurich in numbers of hedgehog rescue centers?

Reviewer 3 Report

Review report for:

Unexpected gene-flow in urban environments: the example of the European hedgehog

Barthel et al. has provided a well-written and highly relevant and interesting manuscript on the genetics of hedgehogs living in an urban environment. An important contribution to our understanding of hedgehog genetics as well as the influence of fragmentation (or lack thereof) in urban environments.

I would suggest to add isolation-by-distance (IBD) to the analyses to make it stronger. This analysis is frequently used in similar publications.

Furthermore, I am a bit concerned about the relatedness, as r < 0.5 seems a too high to exclude close relatedness? However, you do seem to distinguish between close relatedness (Ntotal=143) and unrelated individuals (N=65) during your treatment of results (e.g. 3.2.1). But if “unrelated” is simply defined as r < 0.5 (this was not entirely clear to me L. 193) it is still high?

Comments:

L. 93: I am sure you must have marked the animals (N=250) in some way to distinguish them from each other, but it would be nice if you described how. Also to indicate that the samples were in fact collected from 250 unique individuals.

L.216: Cluster 1 with 29 genotypes, where did these individuals come from? I am sure I can find the GPS locations in the Supplementary material. However, Cluster 2 is almost exclusively Tierpark and Cluster 3 is almost exclusively Treptower Park. So it would be great with some kind of map overview of these clusters since Cluster 1 doesn’t really have a name to assign the location to. As a reader it would be helpful to get a quick and easy overview of the clusters.

L.238: Would it be possible to somehow account for the influence of rescued animals on the results without necessarily have to make the whole analysis again? Because, as you mention, if the animals were released far from their original habitats, they would definitely influence the genetic structure of the population with “fresh blood”. It seems you took the samples when they were admitted for treatment, but it is not clear to me whether they were all released back into the wild again, and if so, where?

- Fig 1: External samples, are they the samples from hedgehog rescue centers and vets? Could you perhaps add this definition to the figure text to ease the interpretation?

- Fig 1: If the external samples are from the individuals taken into care it seems they represent much more distant locations than the “fieldwork” individuals. Here I believe it would be highly beneficial to make an IBD-analyses to account for the effect of distance between samples.

- I would really like to see a figure with a visual overview with a map of the three smaller clusters you have defined? And perhaps their percentagewise share of DNA if it is possible or something to at least indicate the migration data you have (L.225)?

- I suggest you add the very recent publications by Rasmussen et al. on hedgehog genetics to the introduction or discussion?

  • Sophie Lund Rasmussen, Erika Yashiro, Elsa Sverrisdóttir, Kåre Lehmann Nielsen, Mie Bech Lukassen, Jeppe Lund Nielsen, Torben Asp, Cino Pertoldi 2019. Applying the GBS technique for the genomic characterization of a Danish population of European hedgehogs (Erinaceus europaeus). Genetics and Biodiversity Journal (GABJ) vol. 3(2): 78-86. Available at: http://ojs.univ-tlemcen.dz/index.php/GABJ/article/view/688
  • Sophie Lund Rasmussen, Jeppe Lund Nielsen, Owen R. Jones, Thomas B. Berg, Cino Pertoldi 2020. Genetic structure of the European hedgehog (Erinaceus europaeus) in Denmark. PLOS ONE 15(1): e0227205. Available at: https://doi.org/10.1371/journal.pone.0227205

- Regarding your “family clans” it would perhaps be beneficial to include the following points in the discussion: In Rasmussen et al. 2020 we mention the potential influence on genetics of the fact that hedgehogs are promiscuous and have hetero-paternal superfecundation. And also that it is currently unknown whether hedgehogs are able to actively differentiate between kin and non-kin e.g. during the mating season. So this may of course potentially influence the inbreeding/tendency for family clans sharing many genes?

- As a perspective in the discussion you could consider adding ideas or suggestions as to how to measure the influence of landscape features on the genetic results you see (landscape genetics)? In Rasmussen et al. 2020 we tried one method, which of course was just one way to do it and maybe not even the best way?, but perhaps it would be good to address this challenge and inspire/encourage future publications on urban wildlife genetics to include such analyses since you seem to have refrained from making landscape genetic analyses.

I hope you find my comments useful. I enjoyed reading your manuscript. 

All the best,
